# Shedding Light on the Effects of Orienteering Exercise on Spatial Memory Performance in College Students of Different Genders: An fNIRS Study

**DOI:** 10.3390/brainsci12070852

**Published:** 2022-06-29

**Authors:** Shengbin Bao, Jingru Liu, Yang Liu

**Affiliations:** 1School of Physical Education, Shaanxi Normal University, Xi’an 710062, China; baoshengbin2022@126.com; 2College of Physical Education, Northeast Normal University, Changchun 130024, China

**Keywords:** orienteering, spatial memory, college students, gender differences, fNIRS

## Abstract

Objective: To investigate the intervention effect of orienteering exercises on the spatial memory ability of college students of different genders and its underlying mechanism. Methods: Forty-eight college students were randomly screened into experimental and control groups, 12 each of male and female, by SBSOD scale. The effects of 12 weeks of orienteering exercises on the behavioral performance and brain activation patterns during the spatial memory tasks of college students of different genders were explored by behavioral tests and the fNIRS technique. Results: After the orienteering exercise intervention in the experimental group, the male students had significantly greater correct rates and significantly lower reaction times than the female students; left and right dorsolateral prefrontal activation was significantly reduced in the experimental group, and the male students had a significantly greater reduction in the left dorsolateral prefrontal than the female students. The degree of activation in the left and right dorsolateral prefrontals of the male students and the right dorsolateral prefrontals of the female students correlated significantly with behavioral performance, and the functional coupling between the brain regions showed an enhanced performance. Discussion: Orienteering exercises improve the spatial memory ability of college students, more significantly in male students. The degree of activation of different brain regions correlated with behavioral performance and showed some gender differences.

## 1. Introduction

Spatial memory is closely related to way-finding ability and a range of higher cognitive brain activities such as spatial representation, navigation, and decision-making [1]. Since 1948, when Tolman [2] introduced the concept of cognitive maps, the study of memory in animals and humans has become a hot research issue. The formation of memory is the basis for consolidating various cognitive processing activities and plays an essential role in the process of individual survival and development. Spatial memory, as a form of memory, is an indispensable part of human survival, mainly in memory recognition of the environment. It is a memory capacity that is used to express the geographic location or direction of the external environment and mainly includes spatial working memory [3], short-term spatial memory [4] and long-term spatial memory [5]. Previous experimental paradigms for spatial memory have mostly used the Corsi block tapping test [6], the *n*-back task [7] and the spatial span test [8], and experimental materials have mostly used blocks, diamond-shaped pictures, circles and fruit elements. In this study, Google Maps, which is commonly used in daily life, was used as the experimental material to more objectively reflect human spatial-memory ability and explore the value of spatial memory ability in spatial navigation.

It has been shown that spatial memory abilities present gender differences, with males excelling at the behavioral level of spatial memory ability [9,10,11], as evidenced by shorter reaction times and higher accuracy [12]. Some researchers have found through a water-maze experimental-research paradigm that males are considered to be more gifted than females in spatial memory [13,14] and this may influence task performance [15]. However, it has also been shown that there are no differences between males and females in spatial memory abilities [16,17]. Brain cognitive differences in spatial memory between males and females and differences in the benefits of improvement in spatial memory ability through intervention need to be further explored.

A large body of existing research confirms that physical activity also benefits the development of students’ cognitive functions [18,19], including long-term memory [20] and short-term memory [21]. Studies addressing the benefits of interventions for different types of exercise programs have found that aerobic exercise significantly improves memory (immediate vs. delayed memory) [22,23]. Stroth found through an experimental study that aerobic endurance exercise is beneficial in enhancing the visual–spatial memory ability of practitioners [24]. The theory of cortical plasticity suggests that structural and functional improvements in different regions of the brain can be made by means of cognitive training. Aerobic exercise has been found to play a facilitative role in improving brain plasticity and cognitive function through neuroimaging monitoring [25,26], and aerobic exercise enhances spatial memory by improving brain activity and cognitive processing speed in the prefrontal cortex [27,28]. Aerobic exercise can also affect the activation patterns of the brain regions that are involved in higher cognitive functions and their functional connectivity, in addition to its effects on the structure of movement-related brain regions [29,30,31]. In summary, it is evident that aerobic exercise and cognitive training can effectively improve the spatial memory capacity of practitioners and that such changes lie in the improvement of brain cognitive mechanisms.

Orienteering is the crossing of unknown terrain with the help of a map and a compass [32] and it is a high-cognitive aerobic exercise program. Orienteering has been described as a high-cognitive sport with the same unique cognitive value as running a marathon and playing chess at the same time. Compared with other sports, orienteers have certain cognitive advantages in spatial cognition [33]. Specific cognitive training in orienteering enables orienteering practitioners to master effective visual search strategies [34]. A study by Eccles found that those who were good at using attention and spatial memory during orienteering route-planning could complete the race effectively [35]. Eda conducted a study of 20 students with orienteering training before and post-training; the data showed that orienteering exercises had a significant effect on both students’ attention and memory levels [36]. During the orienteering exercise, the practitioner’s perception of orientation, route memory ability, and route decision-making ability were exercised. Therefore, orienteering exercises have some targeting value as a motor intervention to improve spatial memory.

In recent years, fNIRS has been increasingly used in the field of sports, with advances and breakthroughs in tai chi [37], table tennis [38], and aerobics [39]. The prefrontal lobe has been found to play an important role in cognitive processes through fNIRS techniques. It has been suggested that in memory, tasks are associated with structures such as the prefrontal lobe of the cerebral cortex [40,41]. The prefrontal lobe is the area that receives and processes external information from other functional areas of the brain and then integrates brain information such as memory and intention to immediately make a rational plan [42]. The dorsolateral prefrontal lobe (DLPFC) and the ventral lateral prefrontal lobe (VLPEC), as the main functional areas of the PFC, play an important role in brain functions that are related to motor cognition. In this study, the left and right dorsolateral prefrontal lobes and the left and right ventral lateral prefrontal lobes were selected as regions of interest to investigate the underlying mechanisms of the effects of orienteering exercises on spatial memory ability in college students.

To further understand the behavioral performance and brain activation pattern performance of the effects of orienteering exercises on the spatial memory ability of college students of different genders, the following hypotheses were proposed: (1) In the behavioral experiment, there were differences in the correctness and response time in the cognitive task of spatial memory between the experimental and control groups of subjects of different genders before and after the intervention. (2) In the fNIRS functional brain-imaging test, the activation of Oxy-Hb in the prefrontal brain regions was differential between the experimental and control subjects of different genders, showing some neural efficacy, a correlation between the degree of Oxy-Hb activation in different brain regions, and some correlation with cognitive-behavioral performance.

## 2. Materials and Methods

### 2.1. Participants

The Santa Barbara Sense of Direction Scale (SBSOD) questionnaire, proposed by Mary, was used [43]. The revised SBSOD scale, translated by Yuetong Zhao, has good reliability and is a valid tool for measuring individual sense of direction [44]; the scale was administered to first-year students at Shaanxi Normal University. The reliability test (R = 0.87) was also conducted in this study. The final 48 subjects were recruited and randomly divided into 24 subjects of each sex (age 18.16 ± 1.01 years) in the experimental group and 24 subjects of each sex (age 18.63 ± 0.68 years) in the control group using overall random grouping, with no significant difference in age between the experimental and control groups (*p* > 0.05). The selection criteria: (1) a sense of direction assessment score < 3; (2) normal bare eye vision or corrected vision, all right-handed; (3) no traumatic brain injury, psychiatric history, cardiopulmonary disease, rhinitis, depression, etc.; (4) able to know the keyboard key positions well, none of them had participated in similar experiments; (5) no hearing or visual impairment. Subjects will be paid upon completion of the experiment. All 48 subjects recruited in the study signed the experimental informed agreement, and the study was approved by the ethics committee of Shaanxi Normal University.

### 2.2. Experimental Design

The study used a mixed experimental design of 2 (group: experimental group, control group) × 2 (time: pre-test, post-test) × 2 (gender: male, female) with independent variables, including group (2 levels of experimental group and control group); time (2 levels of pre-test and post-test); and gender (2 levels of male and female). The dependent variables were behavioral correctness, reaction time, and oxyhemoglobin in 4 brain regions of the prefrontal lobe of the brain concentration changes. The subjects were asked to fill out a self-administered basic information questionnaire before the experiment to record their basic information, such as gender and age.

### 2.3. Orienteering Intervention Program

The orienteering intervention program for the experimental group was developed after reviewing relevant materials and conducting interviews with relevant experts who engaged in orienteering teaching and training: the intervention lasted 12 weeks, with two interventions per week and 45 min per intervention (Table 1). Heart rate monitoring was implemented in the classroom through the Firstbeat wearable monitoring system [45], and during exercise, the average intensity was controlled in the moderate-intensity heart rate range (120–140 beats per minute) [46]. Guided by deliberate training theory, the intervention consisted of three main modules: mapping exercises and memory punching (simple and complex) training components (The material is shown in Figure 1).

The control group was from badminton and table tennis non-orienteering classes. The experimental group and the control group were basically the same in terms of the number of teaching sessions, teaching time, and teaching intensity, except for the teaching content. To ensure the control of experimental variables, the classes were taught strictly according to the teaching plan.

### 2.4. Spatial Memory Task Test

#### 2.4.1. Experimental Instruments

The experimental instrument is Nirsport 2, a portable near-infrared spectral brain functional imaging system that is used to realize the acquisition of brain blood sample data. The acquired optical data were solved by the modified Lambert Beer law to obtain the blood-oxygen signal data of Oxy-Hb, Deoxy-Hb and Total-Hb [47]. A fNIRS device is more realistic and effective to reflect the neural activation level of the brain with the Oxy-Hb concentration [48]; therefore, in this study, the Oxy-Hb concentration was used to examine the level of brain changes in the subjects, and the sampling frequency was set at 7.8125 Hz.

#### 2.4.2. Test Materials

With the increasing number of experimental studies on spatial memory, experiments using the change perception paradigm all have three phases: memory, interval, and monitoring [49]. However, the paradigm form of experimental design has multiple variations to study participants’ different choices of color, shape, and location between memory items and stimulus items, in order to determine participants’ spatial memory abilities [50,51]. The experiment was conducted on a Dell INS 3891 (Dell Inc, Austin, USA) running E-prime 2.0 (Psychology Software Tools Inc, Sharpsburg, USA). All stimuli were displayed on a Panasonic CF-53 monitor (resolution 1366 × 768, refresh rate 60 Hz, Panasonic Co., Ltd, Kamonichi, Japan), and the experimental material was derived from Google Maps (3D images), which consisted of 800 × 600 pixel images, each consisting of a varying number of buildings, trees, and paths (Figure 2). The subjects were required to use the Dell KB216 keyboard to select which of the WAD options were consistent with the original image; the difference between the options and the original image could be a building, a tree or a path.

#### 2.4.3. Testing Process

First, the subject’s instructions are presented on the screen, and when the subject is ready, they press the space bar to begin the exercise. During the practice phase, feedback in the form of “correct”, “incorrect” or “no response” is displayed in the center of the screen, depending on the subject’s response. Second, in the experimental task, the subject sees a prompt on the computer screen; after the preparation is correct, they must press the space bar to start the test and stare at the cross in the center of the screen for 30 s. As a baseline, a map first appears on the screen, and this must be quickly memorized (the allotted time is 6 s). After 1 s of white screen, 3 maps appear (the presentation time is 6 s). The subject must choose the option that matches the original image among the three options: W, A, D (Figure 3). The test is divided into 3 blocks; one block has 10 trials, and the trial interval is 15 s. As an index of the reaction time and accuracy rate of spatial memory tasks, the absorption and scattering relationship of oxyhemoglobin was recorded with near-infrared when completing the spatial memory task, and the changes of oxyhemoglobin in subjects under the task state were investigated to reflect brain function and other indicators.

### 2.5. Probe Arrangement

The fNIRS photopolar cap consists of 13 sources and 8 detectors, constituting 28 measurement channels. The inter-probe distance was set to 3 cm and, according to the international 10/20 system. According to the existing Anatomical Labeling Systems (LBPA40), to delineate the region of interest (ROI), a total of 4 ROIs were delineated (see Figure 4): left ventral lateral prefrontal (L-VLPFC): Ch1, Ch2, Ch3, Ch4, Ch5, and Ch7; left dorsolateral prefrontal (L-DLPFC): Ch6, Ch8, Ch9, Ch10, Ch11, Ch12, Ch13; right ventral lateral prefrontal (R-VLPFC): Ch23, Ch24, Ch25, Ch26, Ch27, Ch28; and right dorsolateral prefrontal (R-DLPFC): Ch16, Ch17, Ch18, Ch19, Ch20, Ch21, Ch22. The above four ROIs were evenly distributed in the prefrontal lobe using a multi-channel fNIRS data space alignment to MNI space.

### 2.6. Data Analysis

#### 2.6.1. Behavioral Data

To ensure the accuracy of the experimental data, the extreme values with large disparities were removed, and data outside the range of mean ± 3 standard deviations were removed. With the help of SPSS 25.0 software (IBM Inc, Armonk, USA), the measured data were tested for normal distribution. The results of the Kolmogorov –Smirnov test showed that the data in this paper were all greater than the 0.05 threshold and obeyed a normal distribution. In order to illustrate the intervention effect of orienteering exercises on the values of the spatial memory task, a repeated-measures ANOVA of group and time was conducted; later, in order to illustrate the effect of orienteering exercises on students of different genders, the experimental group was subjected to a repeated-measures ANOVA for both time and gender. In case of interactions, a simple effects analysis was performed using Bonferroni’s method; the significance level was set at *p* < 0.05, and the degree of variation in the behavioral data was reported as standard errors.

#### 2.6.2. fNIRS Data

The acquired data were exported directly to the computer for offline analysis after the experiments were completed. In this study, band-pass filtering was used (components greater than 0.1 Hz and less than 0.01 Hz were filtered out) to filter out the effects of heartbeat, respiration, and other factors on the fNIRS data, and a principal components analysis (PCA) was used to remove motion artifacts [47]. The Oxy-Hb values of all trials under the task condition were averaged to obtain the mean value of each channel of the subjects under the task condition, and the Oxy-Hb data of the 6–7 channels contained in the ROIs blood-oxygen signal of ROI [8]. Data were processed as for behavioral data. A Pearson correlation analysis was used to correlate the Oxy-Hb concentrations of each ROI with the correct rate results for college students of different genders before and after the orienteering intervention in the experimental group, and *p* < 0.05 was considered to be statistically significant.

## 3. Results

### 3.1. Behavioral Results

#### 3.1.1. Behavioral Results of Different Groups before and after Exercise Intervention

To explore the effects of orienteering exercises on the spatial memory ability of college students, a 2 (group: experimental group, control group) × 2 (time: pre-test, post-test) repeated-measures ANOVA was used to statistically analyze the correct rate and reaction time of spatial memory tasks before and after the exercise intervention (Table 2, Figure 5).

The results of the repeated-measures variance of correctness showed: a non-significant time main effect [F(1,46) = 1.981, *p* = 0.166, η2 = 0.041]; a non-significant group main effect [F(1,46) = 1.198, *p* = 0.279, η2 = 0.025]; and a significant time and group interaction [F(1,46) = 10.619, *p* = 0.002, η2 = 0.188]. A simple effects test revealed that in the pretest phase, the difference between the experimental and control groups in terms of correctness was not significant [F(1,46) = 2.343, *p* = 0.133, η2 = 0.048]; in the post-test phase, the experimental group had a significantly greater correctness than the control group [F(1,46) = 18.564, *p* < 0.001, η2 = 0.288]; for the experimental group, the post-test correct rate was significantly greater than the pre-test correct rate [F(1,46) = 10.887, *p* = 0.002, η2 = 0.191]; for the control group, the difference between the correct rate of the post-test and the pre-test was not significant [F(1,46) = 1.713, *p* = 0.197, η2 = 0.036].

The results of the repeated-measures variance of the reaction time results showed: a significant time main effect [F(1,46) = 7.083, *p* = 0.011, η2 = 0.133], with a significantly shorter reaction time in the post-test than in the pre-test; a non-significant group main effect [F(1,46) = 1.802, *p* = 0.186, η2 = 0.038]; and a significant time and group interaction [F(1,46) = 8.661, *p* = 0.005, η2 = 0.158]. A simple effects test revealed that: in the pre-test phase, the difference in response time between the experimental and control groups was not significant [F(1,46)= 2.730, *p* = 0.105, η2 = 0.056]; in the post-test phase, the experimental group had a significantly shorter response time than the control group [F(1,46)= 5.660, *p* = 0.022, η2 = 0.110]; for the experimental group, the post-test reaction time was significantly shorter than that of the pre-test reaction time [F(1,46) = 15.704, *p* < 0.001, η2 = 0.255]; for the control group, the difference in reaction time between the post-test and pre-test was not significant [F(1,46) = 0.040, *p* = 0.843, η2 = 0.001].

#### 3.1.2. Behavioral Results of Spatial Memory Ability of College Students of Different Genders in the Experimental Group before and after the Intervention

To explore the differences in the spatial memory ability of college students of different genders before and after the orienteering intervention, a repeated-measures ANOVA of 2 (gender: male, female) × 2 (time: pre-test, post-test) was used to statistically analyze the correct rate and reaction time of the spatial memory tasks of college students of different genders before and after the intervention in the experimental group (Table 3, Figure 6).

The results of the repeated-measures variance of correctness showed: a significant time main effect [F(1,22) = 13.907, *p* = 0.001, η2 = 0.387], with a significantly greater correctness in the post-test than in the pretest; a non-significant gender main effect [F(1,22) = 0.362, *p* = 0.553, η2 = 0.016]; and a significant time and gender interaction [F(1,22) = 4.288, *p* = 0.043, η2 = 0.230]. A simple effects test revealed that the difference in correctness between male students and female students was not significant at the pre-test stage [F(1,22) = 0.664, *p* = 0.424, η2 = 0.029]; at the post-test stage, the correctness of male students was significantly greater than that of female students [F(1,22) = 5.959, *p* = 0.023, η2 = 0.213]; for male students, the correctness of the post-test was significantly greater than the pre-test [F(1,22) = 15.359, *p* = 0.001, η2 = 0.411]; for female students, the difference between the correct rates of the post-test and pre-test was not significant [F(1,22) = 1.835, *p* = 0.189, η2 = 0.077] (Table 3).

The results of the repeated-measures variance at response time showed: a significant time main effect [F(1,22) = 32.648, *p* < 0.001, η2 = 0.597] and significant gender main effect [F(1,22) = 4.932, *p* = 0.037, η2 = 0.183], with male students having a significantly shorter response time than female students; and a significant time × gender interaction effect [F(1,22) = 3.965, *p* = 0.046, η2 = 0.176]. A simple effects test revealed that in the pretest phase, male and female students did not differ significantly in response time [F(1,22) = 0.086, *p* = 0.772, η2 = 0.004]; in the post-test phase, male students had significantly shorter response times than female students [F(1,22) = 9.262, *p* = 0.006, η2 = 0.296]; for male students, post-test response times were significantly shorter than pre-test response times [F(1,22) = 28.282, *p* < 0.001, η2 = 0.562]; for female students, the post-test reaction time was significantly shorter than the pre-test reaction time [F(1,22) = 7.632, *p* = 0.011, η2 = 0.258] (Table 3).

### 3.2. fNIRS Results

#### 3.2.1. fNIRS Results of Different Groups before and after Exercise Intervention

To explore the effect of orienteering exercises on college students’ spatial memory ability, a repeated-measures ANOVA of 2 (group: experimental group, control group) × 2 (time: pre-test, post-test) was used to statistically analyze the fNIRS data of college students’ spatial memory tasks before and after the exercise intervention (Table 4, Figure 7, Figure 8).

The results of the left ventral lateral prefrontal (L-VLPFC) repeated-measures variance showed: a non-significant time main effect [F(1,46) = 1.074, *p* = 0.305, η2 = 0.023]; a non-significant group main effect [F(1,46) = 0.012, *p* = 0.915, η2 = 0.001]; and a non-significant time × group interaction [F(1,46) =0.001, *p* = 0.982, η2 = 0.001] (Table 4).

The results of the right ventral lateral prefrontal (R-VLPFC) repeated-measures variance showed: a non-significant time main effect [F(1,46) = 0.847, *p* = 0.362, η2 = 0.018]; a non-significant group main effect [F(1,46) = 0.001, *p* = 0.991, η2 = 0.001]; and a non-significant time × group interaction [F(1,46) =0.056, *p* = 0.814, η2 = 0.001] (Table 4).

The results of the left dorsolateral prefrontal (L-DLPFC) repeated-measures variance showed: a significant time main effect [F(1,46) = 11.410, *p* = 0.001, η2 = 0.199], with significantly lower activation in the post-test than in the pre-test; a significant group main effect [F(1,46) = 17.691, *p* < 0.001, η2 = 0.278], the time × group interaction was significant [F(1,46) = 12.247, *p* = 0.001, η2 = 0.210]. A simple effects test revealed the following: in the pre-test phase, the activation levels of the experimental and control groups were not significantly different [F(1,46) = 0.280, *p* = 0.599, η2 = 0.006]; in the post-test phase, the activation level of the experimental group was significantly lower than that of the control group [F(1,46) = 32.635, *p* < 0.001, η2 = 0.415]; for the experimental group, the post-test activation was significantly lower than the pre-test [F(1,46) = 23.650, *p* < 0.001, η2 = 0.340]; for the control group, the difference between pre-test and post-test was not significant [F(1,46) = 0.007, *p* = 0.932, η2 = 0.001] (Table 4).

The results of the right dorsolateral prefrontal (R-DLPFC) repeated-measures variance showed: a non-significant time main effect [F(1,46) = 2.661, *p* = 0.110, η2 = 0.055]; a significant group main effect [F(1,46) = 6.557, *p* = 0.014, η2 = 0.125], with significantly lower activation in the experimental group than in the control group; and a time × group interaction was significant [F(1,46) = 11.183, *p* = 0.002, η2 = 0.196]. A simple effects test revealed that: in the pre-test phase, the activation levels of the experimental and control groups were not significantly different [F(1,46) = 0.302, *p* = 0.585, η2 = 0.007]; in the post-test phase, the activation level of the experimental group was significantly lower than that of the control group [F(1,46) = 20.643, *p* < 0.001, η2 = 0.310]; for the experimental group, the post-test activation was significantly lower than the pre-test [F(1,46) = 12.377, *p* = 0.001, η2 = 0.212]; for the control group, the difference between pre-test and post-test was not significant [F(1,46) = 1.467, *p* = 0.232, η2 = 0.031] (Table 4).

#### 3.2.2. Behavioral Results of Spatial Memory Ability of College Students of Different Genders in the Experimental Group before and after the Intervention

In order to explore the differences in the spatial memory abilities of college students of different genders before and after the orienteering intervention, a repeated-measures ANOVA of 2 (gender: male, female) × 2 (time: pre-test, post-test) was used to statistically analyze the fNIRS data of the spatial memory tasks of college students of different genders before and after the intervention in the experimental group (Table 5, Figure 9 and Figure 10).

The results of the left ventral lateral prefrontal (L-VLPFC) repeated-measures variance showed: a non-significant time main effect [F(1,22) = 0.431, *p* = 0.518, η2 = 0.019]; a non-significant gender main effect [F(1,22) = 0.028, *p* = 0.868, η2 = 0.001]; and a non-significant time × gender interaction [F(1,22) =0.049, *p* = 0.827, η2 = 0.002] (Table 5).

The results of the right ventral lateral prefrontal (R-VLPFC) repeated-measures variance showed: a non-significant time main effect [F(1,22) = 0.239, *p* = 0.630, η2 = 0.011]; a non-significant sex main effect [F(1,22) = 0.030, *p* = 0.865, η2 = 0.001]; and a non-significant time × sex interaction [F(1,22) =0.973, *p* = 0.335, η2 = 0.042] (Table 5).

The left dorsolateral prefrontal (L-DLPFC) repeated-measures variance results showed: a significant time main effect [F(1,22) = 22.563, *p* < 0.001, η2 = 0.507], with significantly lower post-test activation than pre-test activation; a non-significant gender main effect [F(1,22) = 0.009, *p* = 0.924, η2 = 0.001]; and a time × gender interaction effect was significant [F(1,22) = 4.331, *p* = 0.049, η2 = 0.164]. A simple effects test revealed that: in the pre-test phase, the activation levels of male students and female students were not significantly different [F(1,22) = 1.530, *p* = 0.229, η2 = 0.065]; in the post-test phase, male students had significantly lower activation than female students [F(1,22) = 4.321, *p* = 0.049, η2 = 0.163]; for male students, the post-test activation level was significantly lower than the pre-test level [F(1,22) = 23.348, *p* < 0.001, η2 = 0.515]; for female students, the difference between pretest and post-test activation was not significant [F(1,22) = 3.568, *p* = 0.072, η2 = 0.140] (Table 5).

The results of the right dorsolateral prefrontal (R-DLPFC) repeated-measures variance showed: a significant time main effect [F(1,22) = 11.448, *p* < 0.001, η2 = 0.507], with significantly lower post-test activation than pre-test activation; a non-significant gender main effect [F(1,22) = 0.941, *p* = 0.343, η2 = 0.041], and a time × gender interaction effect was not significant [F(1,22) =0.128, *p* = 0.723, η2 = 0.006] (Table 5).

### 3.3. Results of Correlation Analysis between Cerebral Blood Oxygen Activation and Correct Rate

A correlation analysis was performed between the Oxy-Hb concentration and behavior (correct rate) at each ROI of the spatial memory task at four levels of gender (male and female) and time (pre-test and post-test) in the experimental group, in order to explore the degree of correlation between activation and behavioral performance.

As seen in Table 6, in the spatial memory task, the ROIs with significant correlations between the fNIRS data results and correct rates in the post-test phase for the male students in the experimental group were L-DLPFC (r = −0.602) and R-DLPFC (r = −0.651); for the female students, the ROIs with significant correlations between the fNIRS data results and correct rates in the post-test phase were R -DLPFC (r = −0.732), and no significant correlations existed between other regions of interest and correct rates (Figure 11).

### 3.4. Analysis of Functional Connectivity Results between Brain Regions

The analysis of brain network connectivity between the four prefrontal regions of interest in the spatial memory task at four levels (group: experimental, control) and (time: pre-test, post-test), respectively, revealed that the L-VLPFC was significantly correlated with the R-VLPFC (r = 0.72) brain regions in the experimental group at the pre-test stage (Figure 12a); the L-VLPFC was significantly correlated with the L DLPFC (r = 0.69) brain regions, and the R-DLPFC correlated significantly with the L-VLPFC (r = 0.71) and L-DLPFC (r = 0.90) brain regions, respectively, in the experimental group at the post-test stage (Figure 12b); in the control group at the pre-test stage, the L-VLPFC correlated significantly with the R-VLPFC (r = 0.59) brain regions (Figure 12c). In the control group, the L-VLPFC correlated significantly with the R-VLPFC (r = 0.69) brain regions at the post-test stage (Figure 12d).

To further investigate the functional connectivity of brain networks between male and female genders in the experimental group before and after the oriented movement intervention, the brain network connectivity between the four prefrontal areas of interest in the spatial memory task was analyzed at four levels (time: pre-test, post-test) and (gender: male, female), respectively, and it was found that the L-VLPFC and R-VLPFC (r = 0.64) brain areas were in the male students., The L-VLPFC correlated significantly with the L-DLPFC (r = 0.64) and the R-DLPFC correlated significantly with the L-VLPFC (r = 0.58) and L-DLPFC (r = 0.94) brain regions, respectively, during the post-test phase (Figure 13a). In the female students, the L-VLPFC correlated significantly with the R-VLPFC (r = 0.94) brain region during the pre-test phase (Figure 13b), and the L-VLPFC correlated significantly with the R-VLPFC (r = 0.64) brain region during the pretest phase (Figure 13b); as well as the VLPFC (r = 0.75) brain regions in the pre-test stage (Figure 13c). In girls, the L-VLPFC was significantly correlated with the L-DLPFC (r = 0.84) brain regions and the R-DLPFC was significantly correlated with the L-VLPFC (r = 0.83) and L-DLPFC (r = 0.65) brain regions, respectively, at the post-test stage (Figure 13d).

## 4. Discussion

### 4.1. Analysis and Discussion of Behavioral Results

From the behavioral results, it can be learned that college students’ spatial memory ability was improved after 12 weeks of orienteering practice, as evidenced by a significant increase in correctness and a decrease in reaction time and enhanced performance. This is consistent with the hypothesis of this study. In the process of orienteering training, initial information or relevant situational features (information such as landscape symbols, map colors, legend notes, etc.) need to be processed, encoded, and stored, and then the relevant responses in memory are extracted to make it into the working memory studio for efficient recognition memory. Effective spatial-memory capacity is key to succeed in orienteering competitions [52]. In sports situations with limited space and time constraints, the need to quickly search for valid information on the field to make appropriate sport decisions requires not only short-term memory processing and storage processes, but also the process of extracting long term memory in the athlete’s brain combined with short term memory to produce cognition [53]. Orienteering is a sport in which performance is measured in terms of time. During training, practitioners need to minimize the time that is spent looking at the map and observing information about the surroundings, which requires memorizing the map. The fewer times the map is looked at, the better the memory effect and the better the performance. In the process of developing training programs, under the guidance of deliberate training theory and through the development of targeted training programs [54], practitioners effectively exercise the spatial memory ability of college students through memory punch cards and memory drawing, and exercise the spatial memory cognitive ability of college students through training methods of different difficulties. The good spatial memory ability of orienteering athletes comes from long-term sports experience and expertise training [55].

This study found that male students were significantly more correct than female students after orienteering exercises, and they had significantly lower reaction times than the female students. Further, male students’ performance was higher than female students’ performance. In previous studies on spatial memory, it was found that males are more adept at using the “survey” strategy, which gives them an advantage in spatial orientation and spatial memory measures based on the map-like representation of landmarks and the spatial relationships between them [56]; on the contrary, females, prefer the “route” strategy, where the order in which landmarks appear is crucial [57]. The experimental material for this study was derived from Google Maps (3D images) and aimed to examine the improvement of the orienteering intervention on the practitioners’ way-finding ability, perception, and memory of maps; therefore, the gender difference in the benefits of the intervention may be because, through practice, male students had better mastered the strategy of the spatial memory of maps and were able to use it well. In addition, it has been found that whether or not gender differences are significant depends to some extent on the difficulty and type of task, and as the task difficulty increases, the working memory load explains these differential results well [58]. The Google Maps material involves more complex cognitive components (shape, color, location), and the present study also reconfirms that task difficulty and type contribute to differences in spatial memory between males and females.

### 4.2. Analysis and Discussion of fNIRS Results

On the spatial memory task, the cerebral blood-oxygen signal was significantly lower on the L-DLPFC and R-DLPFC after the targeted exercise intervention in the experimental group than on the pre-test, with significantly lower activation in the L-DLPFC in male students than in female students; no significant differences were found before and after exercise on both the R-VLPFC and L-VLPFC. In previous studies, conflicting results were found on the cognitive neural aspects of spatial memory training effects, with some studies finding an increase in activation in the corresponding brain regions after training [59]; others showed a decrease in activation in the corresponding brain regions [60]. The present study is consistent with the latter, where cognition enhances the intensity of prefrontal brain functional activity; when cognitive load increases, the better the performance of the area that is associated with the cognitive task, and the lower the activation of its cerebral cortex, showing a negative correlation [61]. This experimental design was limited to the area of interest (PFC) that was related to spatial memory, and the brain mechanisms of college students of different genders during spatial memory tests were investigated. It was found that after the training, a decrease in activation of the prefrontal cortex of the brain was induced, and the efficiency of the use of neural resources in the cerebral cortex was expressed as the neural efficiency of the brain, so the prefrontal cortex of college students may have been exercised in the process of orienteering, causing the activation degree after the orienteering intervention to show a decrease in the phenomenon of enhanced cognitive maps and memory.

It was found that blood oxygen signal concentrations in the dorsolateral prefrontal lobe (DLPFC) were significantly lower in college students following orienteering exercises, and that activation was significantly lower in the left dorsolateral prefrontal lobe in males than in females. The brain area that is activated by spatial memory appears to be variable, and it depends on the length of the memory interval. Shorter intervals will activate the left prefrontal lobe and longer intervals will activate the right prefrontal lobe [62]. The dorsolateral area is a key area for spatial attention, the maintenance of information (memory), and scene memory [63]. According to Cao, cognitive training induced deeper activation of the bilateral DLPFC compared to the group without cognitive training, and long-term higher intensity aerobic exercise interventions were able to enhance the function of the lateral DLPFC [64]. One study investigated the effect of aerobic exercise on brain spatial memory with the help of the fNIRS technique, and aerobic exercise intervention revealed an increase in oxygenated hemoglobin concentration in the right ventral lateral prefrontal lobe [65]. The brain neural mechanisms of spatial memory have also been a hot issue of interest in cognitive neuroscience and neurobiology [66]. 

In the process of orienteering exercises, it is necessary to form representations in the brain through memory of information (such as symbols that are reflected on the map and spatial relationships between the symbols when understanding the map), and then process the representations through spatial imagination operations, finally transforming them into representations that can reflect the three-dimensional real geospatial environment, involving the transformation of two-dimensional maps to realistic three-dimensional maps. The recognition and memorization of route information, point information and symbolic information exercised the brain mechanisms of college students through memory punctuation and memory drawing exercises, which involved the cognitive component of memory in the process of brain cognition, and showed higher neural efficacy after 12 weeks of long-term practice, resulting in differences in brain activation characteristics of spatial memory tasks after the motor intervention.

In the spatial memory task, we analyzed the correlation between behavioral performance and cerebral blood oxygenation, and the male students showed significant correlations between the Oxy-Hb concentrations of the L-DLPFC and R-DLPFC in the prefrontal cortex and correct rates, indicating that after the orienteering motor intervention, their judgments of spatial memory were more associated with the left dorsolateral prefrontal and right dorsolateral prefrontal lobes. The Oxy-Hb concentration of R-DLPFC in the prefrontal cortex of female students was significantly correlated with the correct rate, indicating that after the orienteering intervention, their judgments of spatial memory were more related to the right dorsolateral prefrontal lobe. This correlates with brain areas corresponding to the processing of spatial memory task types, and it has been shown that the prefrontal cortex is an important neural structure involved in memory processing [67]. Functional brain imaging studies have found that the lateral prefrontal cortex is significantly activated when subjects perform spatial memory tasks [68], and that the dorsal region is a key area involved in cognitive dimensions (form, location, order, etc.) and is responsible for decision-making and cognitive control tasks [69,70]; the DLPFC is a functional area involved in multiple cognitive processes [71]. 

Some researchers have suggested that the decline in the cortical activation of brain areas should be combined with the evaluation of functional connectivity or coupling of brain areas to evaluate the neural efficiency of individual brains [72]. Therefore, this study further analyzed the functional connectivity of brain networks among brain regions and found that there was an extremely strong correlation between the degree of brain neural activation and the degree of change in the content of cerebral blood oxygen. One study found that the correlation of brain regions during the task was higher in table tennis than in non-athletes, and the performance of functional coupling among brain regions was enhanced [73,74]. The increased efficiency of brain network transmission in the brain is only an enhancement of local area and backbone-network node connectivity [75,76]. In this study, after 12 weeks of directed exercise practice, the functional brain connectivity between the prefrontal lobes of the brains of college students of different genders was significantly enhanced, probably because long-term aerobic exercise enhances cardiorespiratory function, influences the regulation of cardiac blood uptake activity and cerebral blood circulation, accelerates the exchange of metabolic substances between blood and tissues, and thus reduces the relative concentration of oxygenated hemoglobin in the relevant brain regions, but requires the mobilization of more brain regions. The combined effect covers a wider area. The enhanced connectivity of different brain regions in the prefrontal lobe of the brain by 12 weeks of directed exercise verifies the feasibility of directed exercise intervention for improving the connectivity of brain functions in spatial memory. In conclusion, it can be seen that orienteering exercises have certain effects on the brain structure and function of college students, and long-term orienteering exercises training can improve behavioral performance and spatial memory ability, promote brain neural efficiency, and enhance functional coupling performance between brain regions, which is important to reveal the brain function mechanism of orienteering exercises to promote spatial memory ability and enhance special training performance.

## 5. Conclusions

Integrating behavioral and neuroimaging evidence, the present study found that orienteering exercises can effectively improve spatial memory performance in college students. Further, the prefrontal lobe of the brain showed neural efficacy after orienteering exercise intervention, was conditioned by the type of spatial memory task, showed different activation states in different regions of interest, showed significantly lower activation in the brain regions that were associated with spatial memory tasks, and the degree of activation was correlated with behavioral performance There is a certain correlation and enhanced functional connectivity between different brain regions, showing certain gender differences. This study provides new evidence for uncovering the relationship between motor interventions and spatial memory ability, provides a practical pathway for enhancing spatial memory ability in college students, and provides new evidence for gender differences in the benefits of spatial memory interventions for males and females.

## Figures and Tables

**Figure 1 brainsci-12-00852-f001:**
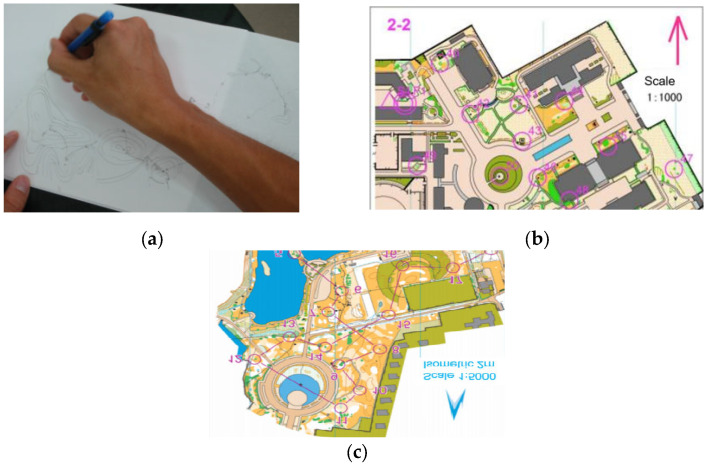
Spatial memory experimental material. (**a**) The purpose of the exercise is to make students proficient in basic information such as symbols, colors and checkpoint description tables in the map; (**b**) aim to improve students’ ability to memorize campus-map route memory; (**c**) aim to improve students’ ability to memorize campus-map route memory.

**Figure 2 brainsci-12-00852-f002:**
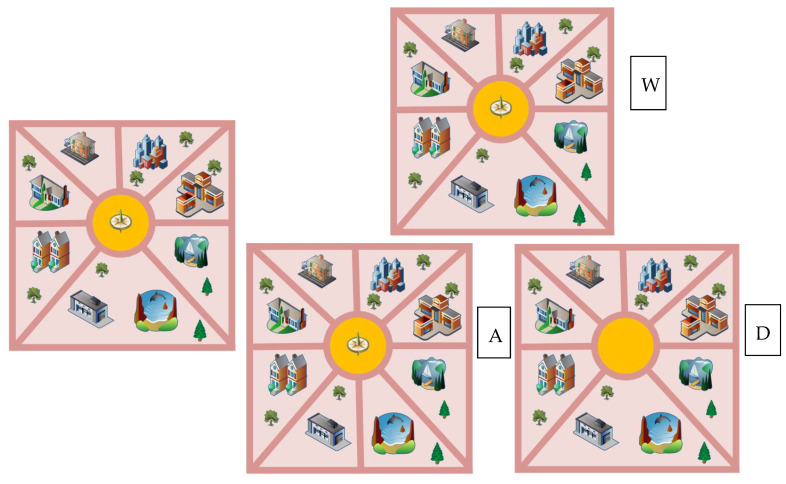
Spatial memory experimental material.

**Figure 3 brainsci-12-00852-f003:**
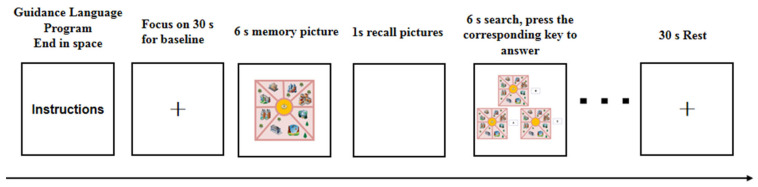
Experimental flow of spatial memory test.

**Figure 4 brainsci-12-00852-f004:**
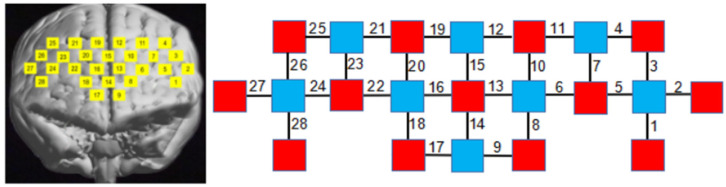
Configuration of the measurement channels in the prefrontal cortex areas. The yellow numbers on the left represent the corresponding detection positions of the photopolar cap in the prefrontal cortical area; the red numbers on the right indicate the emitter (light source); the blue numbers indicate the detector (probe); and the black numbers indicate the established channels.

**Figure 5 brainsci-12-00852-f005:**
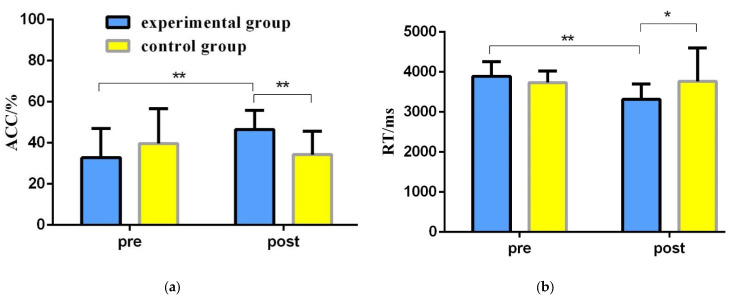
Changes in behavioral results of the experimental group and the control group before and after exercise intervention. (**a**) Changes in the correct rate of spatial memory test before and after the orienteering intervention in the experimental and control groups; (**b**) changes in the response time of spatial memory test before and after the orienteering intervention in the experimental and control groups; blue represents the experimental group, yellow represents the control group. * 0.01 < *p* < 0.05; ** *p* < 0.01.

**Figure 6 brainsci-12-00852-f006:**
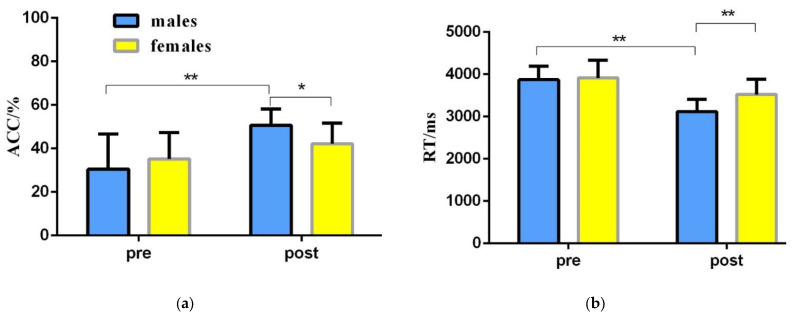
Changes in spatial memory behavior results of college students of different genders before and after orienteering intervention in the experimental group. (**a**) Changes in the correct rate of college students of different genders before and after the O&M intervention; (**b**) changes in the response time of college students of different genders before and after the O&M intervention; blue represents male college students, yellow represents female college students. * 0.01 < *p* < 0.05; ** *p* < 0.01.

**Figure 7 brainsci-12-00852-f007:**
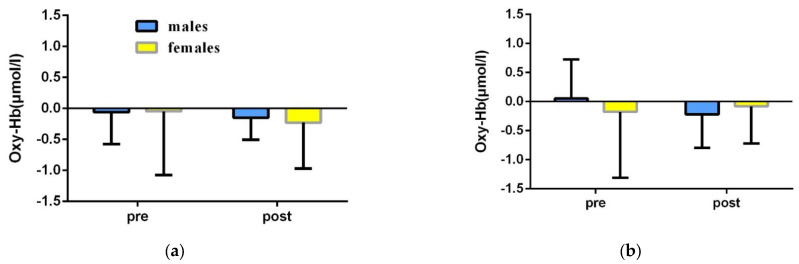
Changes of fNIRS results before and after exercise intervention in experimental group and control group. (Changes in different brain regions before and after directed motor intervention (**a**) L-VLPFC; (**b**) R-VLPFC; (**c**) L-DLPFC; (**d**) R-DLPFC; blue represents the experimental group, yellow represents the control group. * 0.01 < *p* < 0.05; ** *p* < 0.01).

**Figure 8 brainsci-12-00852-f008:**
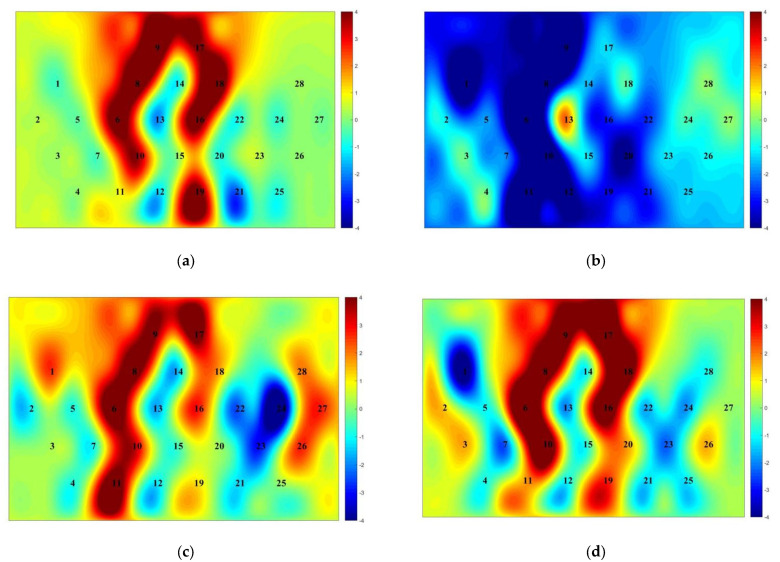
Activation map of prefrontal lobe in experimental group and control group before and after exercise intervention. (**a**) Experimental group pre-test; (**b**) experimental group post-test; (**c**) control group pre-test; (**d**) control group post-test. The numbers represent the 28 channels of the prefrontal lobe and the colors represent the level of activation of the prefrontal lobe, with redder colors representing higher activation and bluer colors representing lower activation.

**Figure 9 brainsci-12-00852-f009:**
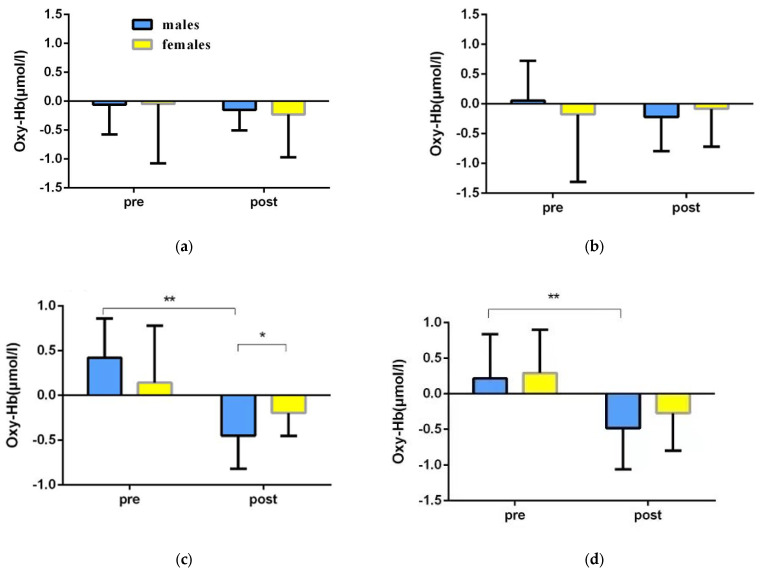
Changes in the fNIRS results of spatial memory of college students of different genders before and after orienteering intervention in the experimental group. Before and after orienteering intervention in the experimental group. (**a**) L-VLPFC; (**b**) R-VLPFC; (**c**) L-DLPFC; (**d**) R-DLPFC) represent the changes of Oxy-Hb concentration in college students of different genders. * 0.01 < *p* < 0.05; ** *p* < 0.01.

**Figure 10 brainsci-12-00852-f010:**
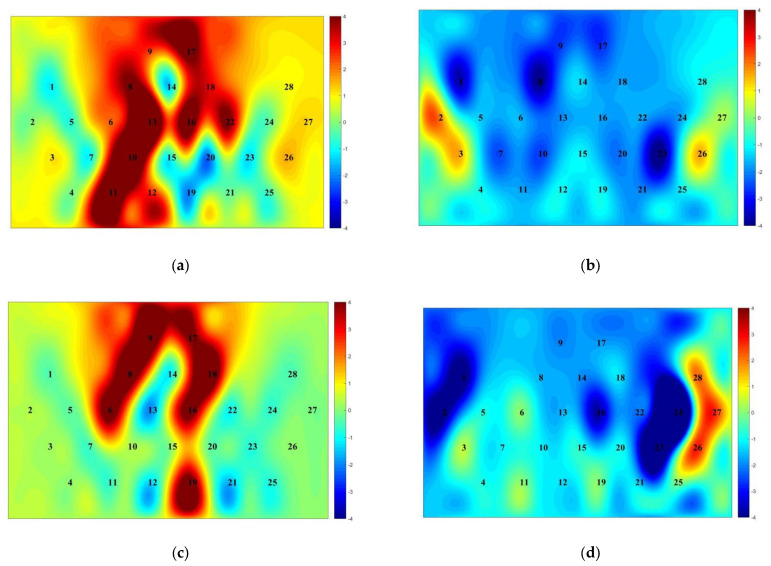
Activation map of the prefrontal lobe of college students of different genders before and after orienteering intervention in the experimental group. (**a**) male pre-test; (**b**) male post-test; (**c**) female pre-test; (**d**) female post-test. The numbers represent the 28 channels of the prefrontal lobe and the colors represent the level of activation of the prefrontal lobe, with redder colors representing higher activation and bluer colors representing lower activation.

**Figure 11 brainsci-12-00852-f011:**
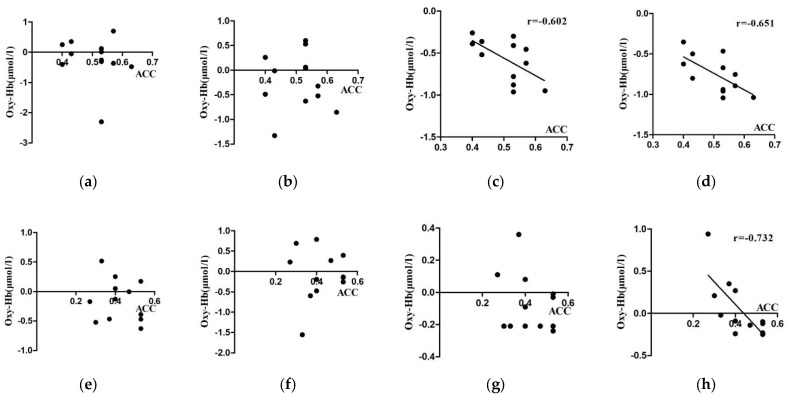
Scatter plot of correlation between fNIRS data and accuracy for the effect of orienteering on gender-specific university students. (Male (**a**): L-VLPFC; (**b**): R-VLPFC; (**c**): L-DLPFC; (**d**): R-DLPFC. Female (**e**): L-VLPFC; (**f**): R-VLPFC; (**g**): L-DLPFC; (**h**): R-DLPFC.)

**Figure 12 brainsci-12-00852-f012:**
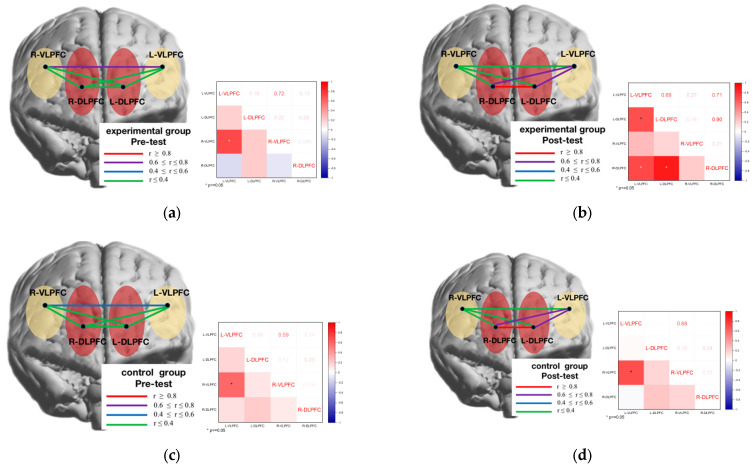
Correlation of brain network functional connectivity in different groups before and after exercise intervention. Brain network connectivity analysis between the four prefrontal regions of interest. (**a**) Pre-test phase in the experimental group; (**b**) post-test phase in the experimental group; (**c**) pre-test phase in the control group; (**d**) post-test phase in the control group. Brain area maps: red area represents dorsolateral prefrontal lobe (DLPFC); yellow represents ventral lateral prefrontal lobe (VLPFC). Correlations between brain areas are represented by connecting lines: red represents r ≥ 0.8; purple represents 0.6 ≤ r ≤ 0.8; blue represents 0.4 ≤ r ≤ 0.6; green represents r ≤ 0.4. Correlation hotspot: red represents positive prior correlation; blue represents negative correlation; and the darker the color, the greater the correlation. * represents *p* < 0.05.

**Figure 13 brainsci-12-00852-f013:**
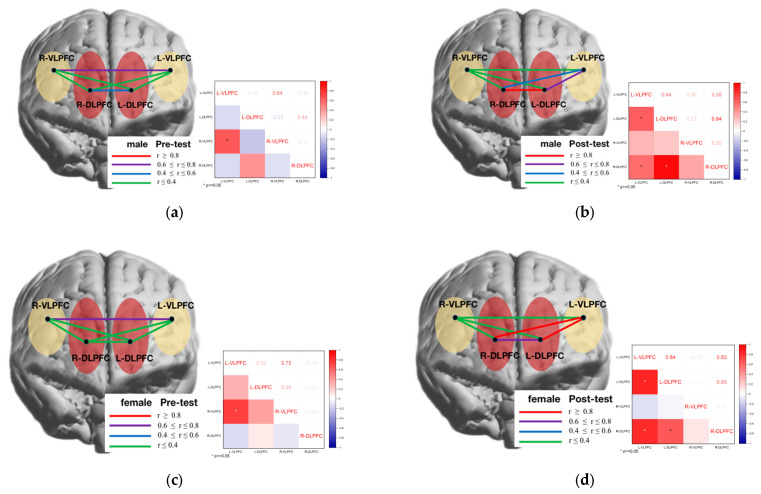
Correlation of functional connectivity of brain networks in different genders before and after directional exercise intervention in the experimental group. (**a**) male pre-test; (**b**) male post-test; (**c**) female pre-test; (**d**) female post-test.

**Table 1 brainsci-12-00852-t001:** Program of orienteering intervention.

Weeks	Course Content	Teaching Content
	Pre-test	Spatial memory test
1–4	Drawing exercise	Purpose of training: to deepen students’ understanding of orienteering map symbols and colors. Practice: master map symbols, colors, contours and checkpoint description sheets. Give students a map with blank areas and ask them to fill in the blanks according to the field situation.
5–8	Exercise in map memorization (simple)	1. Purpose of training: to develop students’ ability to memorize routes on campus. 2. Practice: a point-memory relay race on campus, by setting up point information of different difficulties for memory punching (number of single punches, number of target interference points).
9–12	Exercise in map memorization (complex)	1. Purpose of training: to develop students’ ability to memorize routes in the field. 2. Time: point-memory training in the field, by setting up point information at different levels of difficulty for memory punching (number of single punches, number of target disturbance points).
	Post-test	Spatial memory test

**Table 2 brainsci-12-00852-t002:** Behavioral variance (ANOVA) analysis of the effect of orienteering exercises on the spatial memory ability of college students. (F in Table 2 represents the results of the ANOVA test, where the interaction between exercise intervention time and group repeated-measures ANOVA for correctness was significant, and the interaction between exercise intervention time and group repeated-measures ANOVA for response time was significant).

Source of Variation	ACC (%)		RT (ms)	
	F	η2	F	η2
Group	1.198	0.025	1.802	0.038
Period	1.981	0.041	7.083 *	0.133
Period × group	10.619 **	0.188	8.661 **	0.158

** *p* < 0.01; * 0.01 < *p* < 0.05.

**Table 3 brainsci-12-00852-t003:** Behavioral variance (ANOVA) analysis of the effect of orienteering exercises on spatial memory ability of college students of different genders. (F represents the results of the ANOVA test, where the correct rate of motor intervention time, and gender repeated-measures ANOVA interaction was significant; the gender at response, and motor intervention time repeated-measures ANOVA interaction was significant).

Source of Variation	ACC (%)		RT (ms)	
	F	η2	F	η2
Gender	0.362	0.016	4.932 *	0.183
Period	13.907 **	0.387	32.648 **	0.597
Period × gender	4.288 *	0.230	3.965 *	0.176

** *p* < 0.01; * 0.01 < *p* < 0.05.

**Table 4 brainsci-12-00852-t004:** The fNIRS data variance (ANOVA) analysis of the effect of orienteering exercises on the spatial memory ability of college students.

Source of Variation	L-VLPFC		R-VLPFC		L-DLPFC		R-DLPFC	
	*F*	*η* ^2^	*F*	*η* ^2^	*F*	*η* ^2^	*F*	*η* ^2^
Group	0.012	0.001	0.001	0.001	17.691 **	0.278	6.557 *	0.125
Period	1.074	0.023	0.847	0.018	11.410 **	0.199	2.661	0.055
Period × group	0.001	0.001	0.056	0.001	12.247 **	0.210	11.183 **	0.196

** *p* < 0.01; * 0.01 < *p* < 0.05.

**Table 5 brainsci-12-00852-t005:** The fNIRS data variance (ANOVA) analysis of the effect of orienteering exercises on the spatial memory ability of college students of different genders.

Source of Variation	L-VLPFC		R-VLPFC		L-DLPFC		R-DLPFC	
	*F*	*η^2^*	*F*	*η^2^*	*F*	*η^2^*	*F*	*η^2^*
**Gender**	0.028	0.001	0.030	0.001	0.009	0.001	0.941	0.041
**Period**	0.431	0.019	0.239	0.011	22.563 **	0.507	11.448 **	0.507
**Period × gender**	0.049	0.002	0.973	0.042	4.331 *	0.164	0.128	0.006

** *p* < 0.01; * 0.01 < *p* < 0.05.

**Table 6 brainsci-12-00852-t006:** Correlation results between fNIRS and behavioral (Pearson’s correlation coefficient r).

Period	L-VLPFC		R-VLPFC		L-DLPFC		R-DLPFC	
	Male	Female	Male	Female	Male	Female	Male	Female
**Pre-test**	0.307	0.377	0.171	0.406	−0.035	−0.154	−0.362	0.287
**Post-test**	−0.179	−0.215	−0.037	0.074	−0.602 *	−0.255	−0.651 *	−0.732 **

(* 0.01 < *p*< 0.05; ** *p* < 0.01).

## Data Availability

Data is contained within the article or Appendix A. The data presented in this study are available in Appendix A.

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
