# Peer review of "Shedding Light on the Effects of Orienteering Exercise on Spatial Memory Performance in College Students of Different Genders: An fNIRS Study"

_brainsci, 2022, doi:10.3390/brainsci12070852_

Round 1
Reviewer 1 Report
The Shedding Light on the Effects of Orienteering Exercise on Spatial Memory Performance in College Students of Different Genders: An fNIRS Study, is a very interesting study. It showed the role of orienteering exercises in improving spatial memory. It is written clearly and quite easy to follow.
Several questions regarding the methods:
1. Did the author consider the depression level of the subjects before starting the exercise? Depression might interfere the cognitive ability
2. The study intervention is 12 weeks. Did all subjects fulfill all the schedules? Is there any violation of protocol during the study? Such as lateness of exercise from the schedule
3. From the graph that the author develops, seems that it's not coming from SPSS. Did the author also use Graphpad or other software? Please stated it
There were also some minors in English written that need correction.
Better use male and female instead of male and women
The abbreviation of fNIRS was not written at the beginning of the paper
Section 4 on page 22, should be discussion not result
Line 128. Seems that it is the information on how to write the methods section
Line 216 reflects brain function. and other indicators. (There are 2 full stops, please revised the punctuation)
Line 437 experimental group --> group
Page 23, line 680, there are 3 exercise words, please revised it
Reviewer 2 Report
1) lines 128-135 are template text. Please delete.
2) Table on page 5 should have horizontal dividers between items.
3) Line 182, please explain why that frequency was chosen.
4) Section 2.4.3 (beginning on line 200); it is unclear to whom "you", "he" and "she" are referring. Please change it to 3rd person if used to explain what the subject did and was told.
5)Please make figure 5, 6, 7 have higher resolution.
6) please present all statistics calculations (F, eta squared, p) in a table rather than in large paragraphs. It is difficult for the reader to see and compare.
Author Response
1.Please see the attachment
2.The language description of the article is being embellished.
